# Tumor Growth Ameliorates Cardiac Dysfunction

**DOI:** 10.3390/cells12141853

**Published:** 2023-07-14

**Authors:** Lama Awwad, Rona Shofti, Tali Haas, Ami Aronheim

**Affiliations:** 1Department of Cell Biology and Cancer Science, Ruth and Bruce Rappaport Faculty of Medicine, Technion-Israel Institute of Technology, Haifa 3525422, Israel; lamaaw@campus.technion.ac.il; 2Pre-Clinical Research Authority Unit, Technion-Israel Institute of Technology, Haifa 3200003, Israel; ronasho@proton.me (R.S.); haas@technion.ac.il (T.H.)

**Keywords:** heart failure, cardiac hypertrophy, cardiac fibrosis, tumor, immune system, macrophage polarization

## Abstract

Heart failure and cancer are the deadliest diseases worldwide. Murine models for cardiac remodeling and heart failure demonstrate that cardiac dysfunction promotes cancer progression and metastasis spread. Yet, no information is available on whether and how tumor progression affects cardiac remodeling. Here, we examined cardiac remodeling following transverse aortic constriction (TAC) in the presence or absence of proliferating cancer cells. We show that tumor-bearing mice, of two different cancer cell lines, display reduced cardiac hypertrophy, lower fibrosis and improved cardiac contractile function following pressure overload induced by TAC surgery. Integrative analysis of qRT-PCR, flow cytometry and immunofluorescence identified tumor-dependent M1-to-M2 polarization in the cardiac macrophage population as a mediator of the beneficial tumor effect on the heart. Importantly, tumor-bearing mice lacking functional macrophages fail to improve cardiac function and display sustained fibrosis.

## 1. Introduction

Cardiovascular diseases are the leading cause of mortality and morbidity around the globe, leading to a substantial socioeconomic burden [1]. Heart diseases and cancer together account for more than 40% of deaths in the US. Interestingly, heart failure and cancer share common risk factors such as genetic predisposition, sedentary lifestyle, smoking, alcohol consumption and chemical exposure [2]. While heart failure and cancer have until recently been considered separate diseases, it is becoming evident that they are highly connected and affect each other’s outcome at multiple levels [3,4,5,6]. In particular, several recent studies using multiple mouse models for heart failure demonstrated that heart failure following myocardial infarction [7,8], and even early cardiac remodeling prior to heart failure promotes cell proliferation, cancer cell migration [9,10,11] and metastasis seeding [3,9,12]. In addition, multiple epidemiological studies have suggested that myocardial infarction patients are at a higher risk of developing cancer with a poorer outcome [13], and the same is true for young patients with severe aortic stenosis [12,14].

Whereas tumor proliferation due to heart failure is a well-established phenotype, no information is available on whether and how tumor growth affects the heart. We used clinically relevant breast [15] and lung cancer cells [16] together with cardiac remodeling mouse models. Our previous research indicated that the remodeled heart of tumor-bearing mice displays lower levels of fibrosis compared to mice with no tumors [12]. In the current study, we use a heart-pressure-overload model in the presence or absence of various cancer models to demonstrate that tumor progression suppresses cardiac hypertrophy and fibrosis and improves cardiac contractile function. We go on to uncover that this anti-hypertrophy, anti-fibrosis phenotype occurs via a tumor-dependent activation of macrophage M1-to-M2 polarization. We further show that this switch ameliorates cardiac dysfunction and promotes a beneficial outcome. Thus, the intimate crosstalk between the heart is bidirectional: the remodeled heart promotes cancer progression, and the tumor suppresses cardiac remodeling processes.

## 2. Materials and Methods

### 2.1. Animals

All experimental protocols were approved by the Institutional Committee for Animal Care and Use at the Technion-Israel Institute of Technology, Faculty of Medicine, Haifa, Israel. Approval number IL-157-10-21. All study procedures are in compliance with the guidelines of the NIH Guide for the Care and Use of Laboratory Animals. The number of mice used in each experiment is included in the figure legends.

Age-matched *C57BL/6* and NOD-SCID female and male mice (8–12 weeks old, 20–25 gr) were used. Mice were bred and raised at the Pre-Clinical Research Authority at the Ruth and Bruce Rappaport Faculty of Medicine. Surgery procedures were carried out under isoflurane anesthesia.

### 2.2. TAC Surgery

TAC surgery was performed on female and male mice (8–12 weeks old) as previously described [17]. Aortic constriction was achieved using a 27-gauge blunt needle to create a standardized aorta constriction. The TAC operation was performed by an experienced veterinarian who was blinded to the experimental groups. Mice were followed-up until the endpoint (typically three weeks following TAC operation). Only mice that survived the TAC surgery and reached the endpoint were included in the analysis.

### 2.3. Echocardiography

Mice were anesthetized with 1% isoflurane and kept on a 37 °C heated plate during the procedure. Echocardiography was performed using the micro ultrasound Vevo3100 imaging system (VisualSonics, Fujifilm, Toronto, ON, Canada). Cardiac size, shape and function were analyzed by conventional two-dimensional imaging and M-mode recordings. Maximal left ventricular end-diastolic (LVDd) and end-systolic (LVDs) dimensions were measured in short-axis M-mode images. Fractional shortening (FS) was calculated using the formula: FS (%) = [(LVDd − LVDs)/LVDd] × 100. All values were based on the average of at least 3 measurements for each mouse.

### 2.4. Cancer Cells

The polyoma middle T (PyMT) murine breast carcinoma cell line was derived from the primary tumor-bearing transgenic mice expressing PyMT under the control of the murine mammary tumor virus promotor [15]. These cells were kindly provided by Prof. Tsonwin Hai (Ohio State University, Columbus, OH, USA). The Lewis lung carcinoma (LLC) cells were purchased from the American Type Culture Collection (ATCC). Cells were tested and found to be free of mycoplasma contamination. Cells were cultured in DMEM supplemented with 10% fetal bovine serum (FBS), 1% streptomycin and penicillin at 37 °C, in a humidified atmosphere containing 5% CO_2_.

PyMT and LLC cancer cells (10^6^) were implanted into the mammary fat pad of female mice or subcutaneously into the flanks and orthotopically injected into male mice, respectively. Tumor volume was monitored over time with a caliper and calculated using the formula: width^2^ × 0.5 × length. According to the Institutional Animal Care and Use Committee, the humane endpoint is defined when the tumor size reaches 1500 mm^3^. Tumors of PyMT and LLC cells were maintained for 25 and 20 days, respectively, following the cancer cells injection. Both cancer cells models reached similar tumor weight at the endpoint.

### 2.5. Tail Vein Injection

An experimental pulmonary metastasis assay was carried out by injecting 100 µL PyMT cells into the tail vein (2 × 10^6^ cells/mL, unless otherwise specified). Mice were sacrificed 10 days later.

### 2.6. Fibrosis Staining

Heart tissue was fixed in 4% formaldehyde and left overnight before being embedded in paraffin, serially sectioned at 10-µm intervals, and mounted on slides. Masson’s trichrome staining was performed according to the standard protocol. Images were acquired using a 3DHistech Panoramic 250 Flash III (3DHISTECH Ltd., Budapest, Hungary). Each section was fully scanned. The percentage of interstitial fibrosis was determined by calculating the ratio of the fibrosis area to the total area of the heart section using ImageJ software (version 1.47v) [18]. Each dot represents the mean of the values taken from at least five fields derived from a single mouse.

### 2.7. Hematoxylin and Eosin (H&E) Staining

Lungs were fixed in 4% formaldehyde and left overnight before being embedded in paraffin, serially sliced at 6-μm intervals, and mounted on slides. H&E staining was performed according to the standard protocol. Images were acquired using a 3DHistech Panoramic 250 Flash III (3DHISTECH Ltd., Budapest, Hungary). Each lung section was fully scanned.

### 2.8. Immunofluorescence

Isolated hearts were fixed for 4 h with 4% PFA at 4 °C, rinsed with PBS and incubated in 30% sucrose overnight before being embedded in optimal cutting temperature compound (OCT). Hearts were serially sectioned at 10-µm intervals. Frozen heart sections were stained for F4/80 (Abcam 111101) and counterstained with DAPI, as previously described. Images were acquired with using a 3DHistech Panoramic 250 Flash III (3DHISTECH Ltd., Budapest, Hungary). Each section was fully scanned. For each analysis, 5 fields were randomly chosen and blindly analyzed with ImageJ software (Version 1.47vr) [18].

### 2.9. RNA Extraction and Quantitative Real-Time PCR

Total RNA was extracted from hearts using an Aurum total RNA fatty or fibrous tissue kit (No. 732-6830, Bio-Rad, Hercules, CA, USA), according to the manufacturer’s instructions. Next, cDNA was synthesized from 1000 ng purified mRNA with the iScript cDNA synthesis kit (NO. 170-8891, Bio-Rad). A real-time polymerase chain reaction was performed with QuantStudio3 (Thermofisher Scientific, 5823 Newton Drive, Carlsbad, CA, 92008, USA). Serial dilutions were included for each gene to generate a standard curve. The qRTPCR was performed using the iTaq universal SYBR green supermix (Cat no. 1725124, Bio-Rad). The procedure was as follows: pre-denaturation at 95 °C for 20 min; denaturation at 95 °C for 1 s, 60 °C for 20 min, for 40 cycles. Melt curve at 95 °C for 1 s, 60 °C for 20 min and 95 °C for 1 s. Values were normalized to housekeeping genes GAPDH or Hsp90′s expression levels (as indicated in the figure legends). The primers’ sequences are listed in Supplemental Appendix A.

### 2.10. Hydroxyproline Assay

The hydroxyproline content was assessed as an index for collagen and fibrosis content in cardiac cell lysates using a colorimetric hydroxyproline assay kit (Abcam, USA, Ab222941). The assay was performed according to the manufacturer’s instructions.

### 2.11. Heart Single-Cell Suspension and Flow Cytometry

Heart single-cell suspension and flow cytometry was prepared as previously described [19]. Briefly, hearts were perfused, extracted, finely minced and then incubated with digestive enzymes at 37 °C on a rocking shaker at 50 rpm for 45–60 min. Samples were homogenized with a 40-µm cell strainer. Red blood cells were lysed using ammonium-chloride-potassium (ACK) lysis buffer. Next, samples were centrifuged at 400× *g*, for 5 min, at 4 °C, and the pellet was then suspended with FACS buffer. Cells were immune-stained with the following anti-mouse antibodies: CD45-Alexa Fluor^®^ 700 (BioLegend, 103128, CA, USA), CD11b-PerCP-Cy 5.5 (BioLegend, 101228, CA, USA), F480-PE (BioLegend, 123110, CA, USA) and CD206-BV421 (BioLegend, 141717, CA, USA). Cells were incubated (30 min, 4 °C) with the antibody mixture in a staining buffer (PBS containing 1% bovine serum albumin and 0.05% sodium azide) and then washed twice with a staining buffer. Cells were acquired using a LSRFortessa flow cytometer (BD Biosciences, NJ, USA). The data were analyzed using FlowJo V.10 software (FlowJo, Ashland, OR, USA).

### 2.12. Blood Serum

Blood was obtained from the facial vein with a 4-μm sterile Goldenrod Animal Lancet (MEDIpoint, Inc., Mineola, NY, USA). Blood was collected and allowed to clot at room temperature for 2 h, followed by 15 min of centrifugation at 2000× *g*. Serum was immediately aliquoted and stored at −20 °C for future use.

### 2.13. Cytokine Array

Serum was obtained from 4 groups: naïve, PyMT-bearing, TAC-operated, PyMT-bearing and non-bearing mice and applied (75 μL) to Proteome Mouse XL Cytokine Array filters (Cat. ARY028, Norcross, GA, USA) according to the manufacturer’s instructions. The signal corresponding to each factor in the array was quantified by TotalLab software analysis (Version 2.2). The level of expression of each protein on the array was calculated relatively to the value obtained for the positive control.

### 2.14. Macrophage Depletion

Macrophage depletion was performed using the macrophage depletion kit using liposomes containing either clodronate or saline (Encapsula Nanosciences LLS, USA, SKU# CLD-8901). Liposomes were injected IP according to the manufacturer’s instructions. The clodronate treatment was initiated together with cancer cell implantation, followed by injections twice a week until mice sacrifice at a humane endpoint.

### 2.15. Cell Size Analysis

Heart tissue was fixed in 4% formaldehyde overnight, embedded in paraffin, serially sectioned at 10 µm intervals and then mounted on slides. Sections were stained following deparaffinization with wheat germ agglutinin FITC, conjugated (Sigma Aldrich Cat #L4895, MA, USA) and diluted to a 1:30 phosphate buffered saline (PBS). Sections were washed three times with PBS and mounted in fluorescence mounting medium (Dako, S3023). Images were acquired by using 3DHistech Panoramic 250 Flash III (3DHISTECH Ltd., Budapest, Hungary). Quantification of the cell size was performed with ImageJ software analysis (Version 1.47vr). Five fields on each slide were photographed. Unstained areas were then identified and segmented using Image Pro Plus software. In each stained area, the mean cell perimeter and area was calculated, and the number of cells was measured.

### 2.16. Statistics

Data are presented as mean ± SEM. All mice were included in each statistical analysis, unless euthanized for humane reasons before the experimental endpoint. Animals were chosen in a randomized fashion for each experimental group. Statistical significance was determined by comparing the two means using a Student’s *t*-test. Statistical analysis was performed with GraphPad Prism 8 software (La Jolla, CA, USA). Values of *p* < 0.05 were accepted as statistically significant.

## 3. Results

### 3.1. Tumor Growth Reduces Cardiac Dysfunction

To examine whether and how cancer affects cardiac remodeling, we used transverse aortic constriction (TAC) surgery to study the pressure overload-dependent cardiac remodeling response in the presence or absence of a solid tumor (orthotopic breast cancer model polyoma middle T (PyMT) [15]). PyMT cells (10^6^) were implanted in the mammary fat pad of 8–12-week-old *C57/Bl6* syngeneic mice, and a TAC operation was performed 5 days later (Figure 1A). TAC-operated, non-tumor-bearing mice served as a control. Heart contractile function was assessed by echocardiography prior to sacrifice, and the fractional shortening (FS%), ejection fraction (EF%) and heart wall thickness were calculated. Our results show that non-tumor-bearing mice had low contractile function, which reached an average of 23.03% ± 3.16 FS%. In contrast, tumor-bearing mice displayed significantly higher contractile function, reaching almost normal FS levels, 28.14% ± 2.6 (Figure 1B). Moreover, this was also observed in the ejection fraction and wall thickness parameters (Appendix A).

Mice were sacrificed at a humane endpoint (25 days after PyMT injection), and their hearts were collected. The hearts derived from the PyMT-bearing group were significantly smaller in size and had a more elliptical shape compared to their non-tumor-bearing counterparts (Figure 1C). Furthermore, the ratio of the ventricular weight to the body weight (VW/BW), a parameter reflecting the extent of cardiac hypertrophy, was lower in the PyMT-bearing mice than in the non-tumor-bearing mice (Figure 1D), a finding consistent with improvement of cardiac contractile function (FS). As the average mouse body weight in the two groups was similar, the increased ratio in the non-tumor-bearing mice represents a net increase in the ventricles’ weight. This decrease in VW/BW ratio was accompanied by a decrease in cardiomyocytes size in the tumor-bearing mice (Appendix A). Consistently, the non-tumor-bearing mice also displayed higher expression levels of brain natriuretic peptide (BNP), a hallmark of cardiac remodeling, compared to the PyMT-bearing mice (Figure 1E).

To measure the extent of fibrosis, which is associated with cardiac dysfunction, we stained heart sections with Masson’s trichrome collagen stain. Representative images and the quantification of fibrosis revealed significantly lower levels of fibrosis in the PyMT-bearing mice compared to those in the non-tumor-bearing cohort (Figure 2A,B). This accumulation of cardiac collagen was consistent with the lower expression levels of two fibrosis hallmark markers (CTGF and Col1α1) in PyMT-bearing mice compared to the non-tumor-bearing mice (Figure 2C). Moreover, a colorimetric analysis of collagen in cardiac lysates showed that non-tumor-bearing mice display higher collagen levels in their hearts compared to their PyMT-bearing counterparts (Figure 2D).

Next, we examined whether the tumor’s beneficial effect on the heart is specific to PyMT cancer cells. Towards this end, Lewis lung carcinoma (LLC) cells were implanted in 8-week-old male mice and TAC-operation was performed 5 days later (Appendix A). Consistent with the results obtained with the PyMT cancer cells, LLC-bearing mice showed higher contractile function (Appendix A), a lower VW/BW ratio (Appendix A), decreased levels of hypertrophy markers (Appendix A), and reduced expression of fibrosis markers in the heart (Appendix A) compared to the non-tumor-bearing group. Cardiac remodeling in the TAC-operated, LLC-bearing and PyMT-bearing mice was indistinguishable from the two respective non-TAC-operated cohorts (controls). Thus, both breast- and lung-derived malignant tumors display a similar reduction in the cardiac remodeling phenotype and improved cardiac contractile function following TAC.

### 3.2. Breast Cancer Malignant Metastasis Confers a Similar Cardiac-Dysfunction Protection Phenotype

Since a more clinically relevant scenario in cancer patients is metastatic lesions, we examined whether the observed tumor-protective effect on cardiac remodeling is also seen in a metastatic experimental model. Due to humane endpoint considerations, mice were first TAC operated, followed by one group receiving a tail vein injection of PyMT cancer cells ten days after surgery and the other not (control). Mice were sacrificed 10 days after the injection (i.e., 20 days following TAC surgery) (Figure 3A). The lungs of the PyMT-injected mice were heavier than those of the non-injected mice (Figure 3B). Hematoxylin and eosin (H&E) staining revealed significant malignant lesions within the lungs of the PyMT-injected mice (see representative image in Figure 3C). Echocardiography analysis of mice’s heart contractile function found that the PyMT-injected mice had a higher FS% than mice with no tumor (Figure 3D). The PyMT-injected mice likewise exhibited a reduced VW/BW ratio (Figure 3E). Collectively, our findings indicated that the tumor beneficial effect occurs also in a pulmonary metastatic cancer model.

### 3.3. The Tumor Beneficial Effect Is Not Mediated by the Adaptive Immune System

One of the mechanisms suggested to mediate tumor-heart crosstalk is the immune system [8]. To examine the possibility that the adaptive immune system mediates the tumor beneficial effect on the heart, NOD-SCID mice lacking both B and T cells and with decreased NK cell functionality were TAC-operated in the presence or absence of a PyMT tumor (Appendix A). TAC-operated, non-tumor-bearing NOD-SCID mice displayed lower contractile function (Appendix A) and a higher VW/BW ratio compared to TAC-operated, PyMT-bearing mice (Appendix A). Consistently, a qRT-PCR analysis showed that the PyMT-bearing mice had decreased expression levels of hypertrophic gene markers (Appendix A) and lower levels of fibrosis markers (Appendix A). Collectively, these results clearly exclude the involvement of T and B cells in mediating the tumor-dependent beneficial effect on cardiac hypertrophy and fibrosis.

### 3.4. M1 and M2 Macrophage Polarization in the Heart of Tumor-Bearing Mice

Next, we examined the role of the innate immune cells, and specifically cardiac macrophages, due to their well-documented role in tissue repair and regeneration and in heart disease deterioration [20]. The expression levels of gene F4/80, a marker of macrophages, determined by qRT-PCR of mRNA derived from the hearts and tumors. In the tumors, the F4/80 gene marker was higher in TAC-operated mice than in those of non-TAC-operated mice (Appendix A). Additionally, F4/80 expression levels were higher in the hearts of PyMT-bearing mice vs. mice with no tumor (Figure 4A), a finding validated using immunofluorescence staining for F4/80 in heart sections (Appendix A). These results indicate an increase in the total number of macrophages both in the heart and in the tumor of mice groups that had both diseases.

Macrophages can be split into two main functionally distinct populations, pro-inflammatory (M1) and anti-inflammatory (M2) [21]. Our examination of the expression levels of gene markers specific for M1 (iNOS) and M2 (CD206) macrophages revealed an increase in the tumor in the M1 gene marker and a decrease in the M2 marker (Appendix A). Interestingly, an opposite effect was observed in the hearts: the M1 marker decreased while the M2 markers (CD206, Arg1 and CD163) increased in TAC-operated, PyMT-bearing mice compared to non-tumor-bearing mice (Figure 4B–E). The anti-inflammatory switch toward M2 macrophages in the heart was further supported by a flow cytometry analysis (Appendix A). Consistently, a similar M1-to-M2 polarization occurs in NOD-SCID mice hearts as well (Supplemental Appendix A).

To explore whether the increase in M2 macrophages is a local (conversion of resident M1 macrophages) or systemic event (i.e., M2 macrophages recruited from the circulation), we measured the expression of the M1 and M2 polarization-inducing cytokines INFγ, TNFα, IL-13 and CCL2 [20,22]. This analysis revealed that while in the heart of PyMT-bearing mice, INFγ and TNFα are at low levels; IL-13 and CCL2 are elevated in the PyMT tumor (Figure 4F–I). This finding is further supported by our measurement of cytokines in the serum of TAC-operated mice in the absence or presence of a PyMT tumor compared with serum derived from naïve mice (Appendix A). Namely, multiple cytokines that promote the M1-to-M2 switch (including IL-5, IL-10, IL-17, IL-22) were higher in the serum of PyMT-bearing mice independent of the presence and absence of TAC, indicating the involvement of tumor properties in inducing this switch. These results can explain the cardiac macrophages phenotype and suggest that local macrophages undergo M1-to-M2 polarization. Nevertheless, M2 macrophage recruitment from the circulation cannot be excluded.

### 3.5. Tumor Growth in Macrophage-Depleted Mice Fails to Promote a Cardiac Beneficial Outcome

To validate the role of macrophages in the tumor-induced cardiac beneficial effects, we performed a TAC-operation in the presence or absence of a PyMT tumor in mice treated with liposomes containing either phosphate-buffered saline (PBS) or clodronate (Figure 5A). Clodronate depletes macrophage populations via endocytosis-induced macrophage death. The experimental timeline was similar to those of the previous experiments. The tumor volume in both experimental cohorts was alike throughout the experimental timeline (Figure 5B), thus excluding any effect of the clodronate treatment on tumor growth. In contrast, the clodronate treatment resulted in a massive depletion of macrophages in the heart, as determined by both flow cytometry and qRT-PCR (Appendix A). Importantly, the tumor-induced cardiac beneficial effects were completely abrogated in the TAC-operated, clodronate-treated group. Furthermore, in mice with a decreased macrophage count, tumor growth failed to improve cardiac contractile function (Figure 5C), hypertrophy (Figure 5D–F), and fibrosis (Figure 5G,H). A schematic representation of the interplay between the remodeled heart and tumor in light of our findings is provided in Figure 6.

## 4. Discussion

While previous studies have suggested that cardiac remodeling promotes cancer progression, here we show that tumor-bearing mice display a reduced cardiac remodeling phenotype. The suppression of cardiac hypertrophy and fibrosis in the presence of a solid tumor is largely mediated by the recruitment and polarization of specific M2 macrophage populations, resulting in the amelioration of cardiac function and an anti-fibrotic outcome. Our previous study suggested that tumor-bearing mice display improved cardiac outcome [12] compared to cardiac remodeling processes in non-tumor-bearing mice [23]. In this study, we demonstrate that tumor growth does indeed affect cardiac remodeling, confirmed using breast- and lung-cancer cell lines and a pulmonary metastasis model. The noted benefits to the heart were suppressed cardiac remodeling and, consequently, reduced hypertrophy and fibrosis and increased contractile function.

This study further identifies macrophages as mediators of the cancer-induced beneficial cardiac phenotype, whereas the involvement of B and T cells was excluded. Notably, we observed a significant increase in the number of macrophages in both the heart and the tumor. While the balance towards M1 macrophages in the tumor increased following TAC, the heart displayed an increase in the M2 macrophage population. Notably, an increase in M1 macrophages was also observed in non-tumor-bearing mice following TAC, a rise that may be due to the involvement of inflammatory macrophages in heart dysfunction acceleration and deterioration. The importance of macrophages for the cardiac beneficial effects was further demonstrated by a macrophage depletion protocol using clodronate containing liposomes.

Using qRT-PCR with mRNA-specific M1 and M2 markers, FACS analysis and immuno-fluorescence staining, we demonstrated that tumor-bearing mice lacking macrophages fail to display the cardiac beneficial phenotype following TAC. This suggests that the tumor beneficial effects are mainly mediated through M1-to-M2 cardiac macrophage polarization [20,24,25].

Whether the observed M1-to-M2 skewing in the heart represents cardiac resident or recruited macrophages is yet to be determined. We show that the synthesis of the M2-polarizing cytokine IL-13 and CCL2 are elevated in the heart. This is consistent with a previous study showing that IL-13 induces M2 polarization, leading to improved cardiac function and reduced heart injury in a viral myocarditis mouse model [26]. Moreover, CCL2 has been shown to play an important role in preventing left ventricular dysfunction and remodeling after myocardial infarction [22]. A cytokine analysis in the serum revealed elevated levels of multiple M1-to-M2 polarizing factors in tumor-bearing mice independent of TAC surgery. On the one hand, cytokines IL-5 [27], IL-10 [28], IL-17 [29] and IL-22 [30] are known to induce M2 polarization via several cellular pathways, including NF-kB, STAT3 and heat-shock proteins. On the other hand, in the tumor, M1-polarizing cytokine INFγ is prominent [20]. Importantly, the accumulation of M2 macrophages in the heart is consistent with their repair and regeneration capacity [20,25,31]. The fact that M2 macrophages have a repair capacity strongly suggests that their therapeutic potential is mediated by the growing tumor.

The experimental set up with tumor-bearing mice does not address whether the tumor can reverse existing cardiac fibrosis or suppress cardiac remodeling processes. Nevertheless, the use of the experimental pulmonary metastasis model, in which mice were first TAC-operated and subsequently cancer cells were injected, supports the suppression and reversal of existing cardiac remodeling processes. This is further supported by our previous study in which we implanted cancer cells 10- and 45-days following TAC and subsequently observed the beneficial cardiac effects [12]. Interestingly, a recent study from our lab, using MDX mice, an experimental mouse model for Duchenne muscular dystrophy (DMD) disorder, revealed that tumor growth inhibits de novo synthesis as well as dissolving existing fibrosis [32]. The ability to dissolve existing fibrosis is clearly an important line of research, since it is considered a direly unmet clinical need [33]. Fibrotic diseases are estimated to account for up to half of all deaths in the developed world [34,35]. The pathophysiologic principles of fibrosis are found in several fibrotic diseases, involving multiple organs such as the liver [36], heart [37,38], skeletal muscles [39], kidney [35] and lungs [40].

It is currently challenging to assess cancer-dependent cardiac effects in patients, mainly because once heart failure patients are diagnosed with cancer, the cardiotoxicity that follows cancer therapy masks the putative cardiac beneficial effects.

In conclusion, heart disease and cancer exhibit an intimate interaction that can be regarded as a reciprocal interplay: cardiac remodeling promotes cancer progression, and the growing tumor, in turn, improves cardiac outcome. This new, unexpected link between cardiac remodeling and tumor progression is highly intriguing given the current lack of knowledge in this niche. It is clear that tumor implantation is not a considerable treatment for heart failure and fibrosis, nevertheless, identification of polarizing cytokines inducing an M1 to M2 switch can be used as a possible therapy. Cancer paradigms can, hence, be harnessed to treat existing heart failure and fibrosis diseases as well as improve patient survival rates.

## 5. Conclusions

This paper highlights the intimate link between heart failure and cancer by demonstrating that proliferating cancer cells in mice models dampen a heart phenotype induced by TAC, an amelioration driven by cardiac macrophages shifting from an inflammatory to an anti-inflammatory state. Harnessing cancer paradigms that are involved in tumor-dependent improved cardiac outcome may provide novel therapeutic strategies for cardiovascular diseases.

## Figures and Tables

**Figure 1 cells-12-01853-f001:**
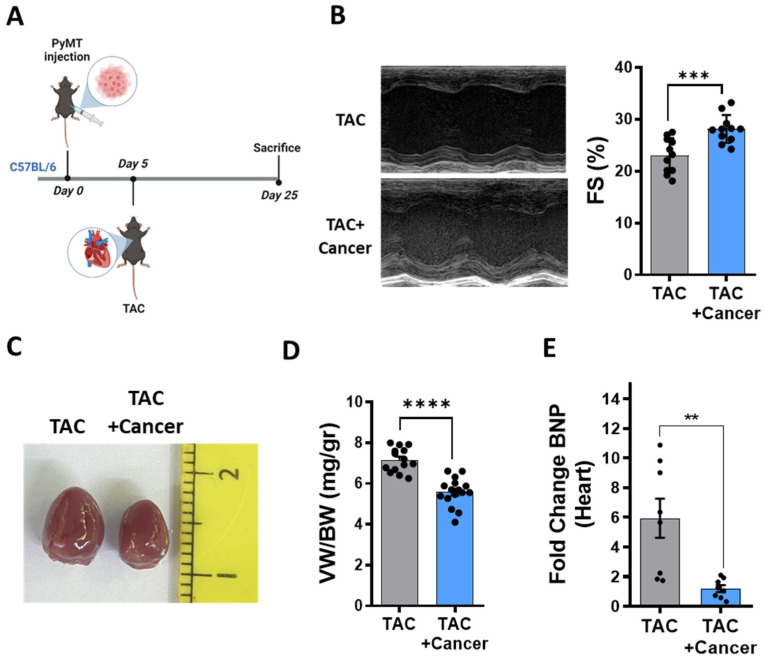
Tumor growth imposes a cardiac beneficial phenotype following TAC-operation in female mice. (**A**). Schematic diagram of the experimental timeline. (**B**). The fractional shortening (FS; right) of TAC-operated, PyMT-bearing (*n* = 11) and non-cancer-bearing mice (*n* = 11), calculated from the respective echocardiography data (left). Calculated based on parameters in Appendix A. (**C**). Representative images of the hearts of a TAC-operated, PyMT-bearing and non-cancer-bearing mice at endpoint (25 days after PyMT injection). (**D**). The ventricular weight/body weight ratio (VW/BW mg/gr) of TAC-operated, PyMT-bearing (*n* = 16) and non-cancer-bearing mice (*n* = 14). (**E**). Level of the hypertrophic hallmark gene marker BNP measured using qRT-PCR with cDNA derived from heart mRNA of TAC-operated, PyMT-bearing (*n* = 8) and non-cancer-bearing mice (*n* = 8), normalized to the GAPDH housekeeping gene. The results are presented as mean ± SEM, Student’s *t*-test ** *p* < 0.01, *** *p* < 0.001, **** *p* < 0.0001. In (**B**,**D**,**E**), each dot represents one mouse.

**Figure 2 cells-12-01853-f002:**
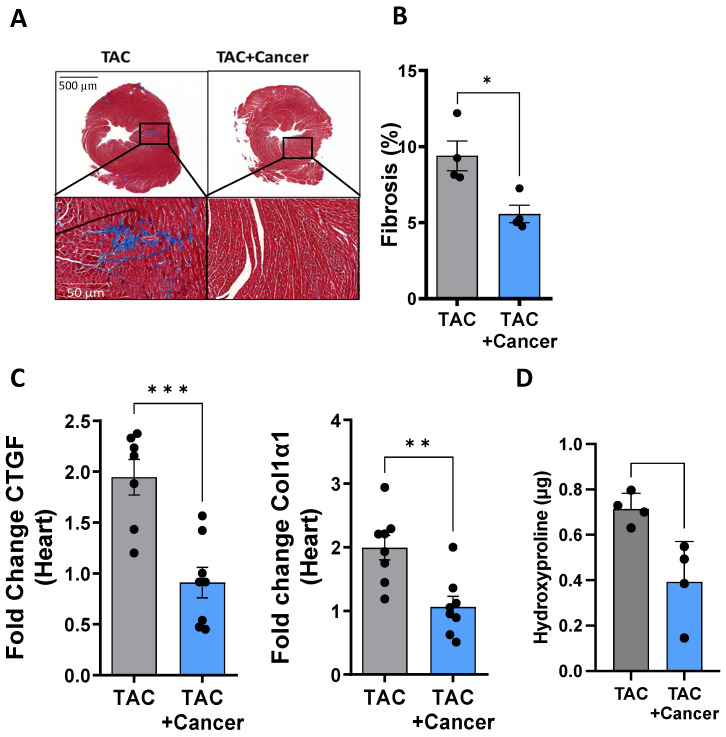
Tumor growth suppresses fibrosis in TAC-operated female mice. (**A**). Representative paraffin-embedded heart sections of TAC-operated, PyMT-bearing and non-cancer-bearing female mice stained with Masson’s trichrome to visualize fibrosis. Scale bar: 500 µm (top) and 50 µm (bottom). (**B**). Quantification of the level of fibrosis (%) by Masson’s trichrome staining in TAC-operated, PyMT-bearing (*n* = 4) and non-cancer-bearing female mice (*n* = 4), at least 5 random fields were chosen per mouse. (**C**). Fibrosis hallmark gene markers CTGF and Col1a1 in TAC-operated, PyMT-bearing (*n* = 8) and non-cancer-bearing (*n* = 8) female mice, measured using qRT-PCR, normalized to the GAPDH housekeeping gene. (**D**). Hydroxyproline levels in the heart lysates of TAC-operated, PyMT-bearing (*n* = 4) and non-cancer-bearing (*n* = 4) female mice. The results are presented as mean ± SEM, Student’s *t*-test. * *p* < 0.05, ** *p* < 0.01, *** *p* < 0.001. In (**B**–**D**), each dot represents one mouse.

**Figure 3 cells-12-01853-f003:**
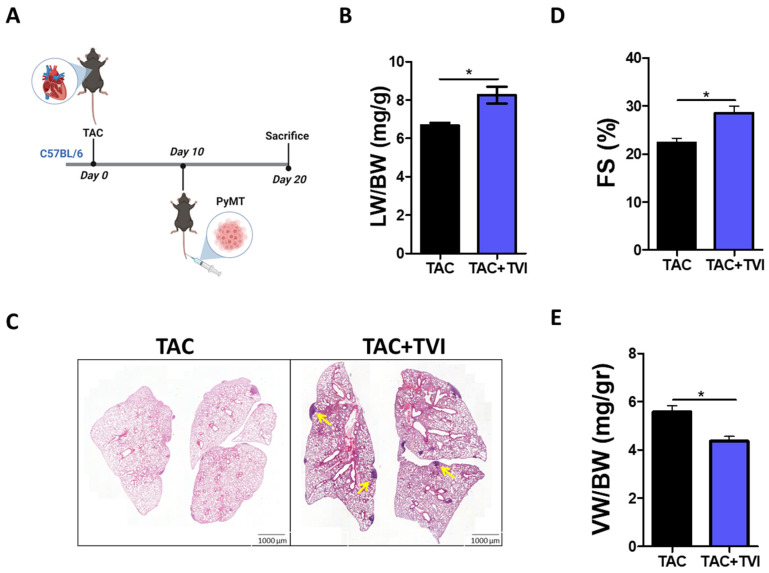
The protective effect on cardiac remodeling is observed also in metastatic cancer model. (**A**). Schematic diagram of the experimental timeline. (**B**). The lungs weight/body weight ratio (LW/BW mg/gr) of TAC-operated, PyMT-injected and non-injected mice. (**C**). Representative images of the H&E staining of lungs of TAC-operated, PyMT-injected and non-injected mice. Yellow arrows indicate some of the metastatic lesions. Scale bar 1000 µm. (**D**). The fractional shortening (FS) of TAC-operated PyMT-injected and non-injected mice at endpoint. Calculated based on parameters in Appendix A. (**E**). The ventricular weight/body weight ratio (VW/BW mg/gr) of TAC-operated, PyMT-injected and non-cancer-bearing mice. The results are presented as mean ± SEM, Student’s *t*-test * *p* < 0.05. (*n* = 3 in each group).

**Figure 4 cells-12-01853-f004:**
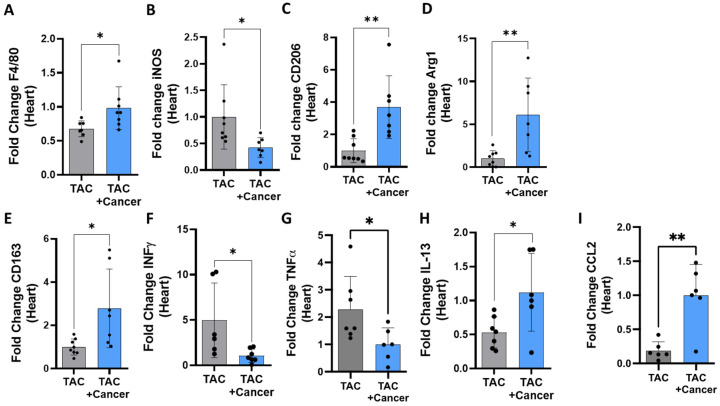
Tumor growth induces cardiac macrophage polarization. (**A**–**I**) The levels of macrophage M1 and M2 hallmark gene markers F4/80 (**A**), iNOS (**B**), CD206 (**C**), ARG1 (**D**), CD163 (**E**), INFγ (**F**), TNFα (**G**), IL-13 (**H**) and CCL2 (**I**) in the hearts of TAC-operated, PyMT-bearing (*n* = 8) and non-cancer-bearing (*n* = 8) mice, measured using qRT-PCR, normalized to the GAPDH housekeeping gene. The results are presented as mean ± SEM, Student’s *t*-test. * *p* < 0.05, ** *p* < 0.01. Each dot represents one mouse.

**Figure 5 cells-12-01853-f005:**
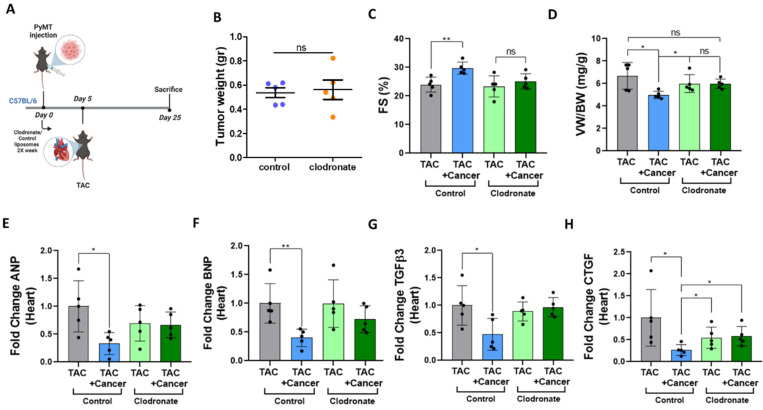
Cardiac beneficial phenotype in PyMT-bearing mice is abrogated in macrophage-depleted mice. (**A**). Schematic diagram of the experimental timeline. Clodronate or PBS (control) liposomes were intraperitoneal (IP) injected at day 0 together with PyMT cancer cell implantation or not (control), and on day 5, a TAC was performed. Subsequently, clodronate/PBS liposomes were injected twice a week until sacrifice. (**B**). The tumor weight of clodronate- and control liposome-injected groups at the endpoint. (**C**,**D**). The fractional shortening (FS), (**C**), calculated from echocardiography data, calculated based on parameters in Appendix A and ventricular weight/body weight ratio (VW/BW), (**D**) in the four mouse cohorts. (**E**,**H**). The heart expression levels of hypertrophy markers ANP (**E**) and BNP (**F**) and fibrosis markers TGFβ3 (**G**) and CTGF (**H**), measured using qRT-PCR and normalized to the HSP90 housekeeping gene. The results are presented as mean ± SEM, Student’s *t*-test. * *p* < 0.05, ** *p* < 0.01, non specific (ns). In (**B**–**H**), each dot represents one mouse. (*n* = 5 in all groups).

**Figure 6 cells-12-01853-f006:**
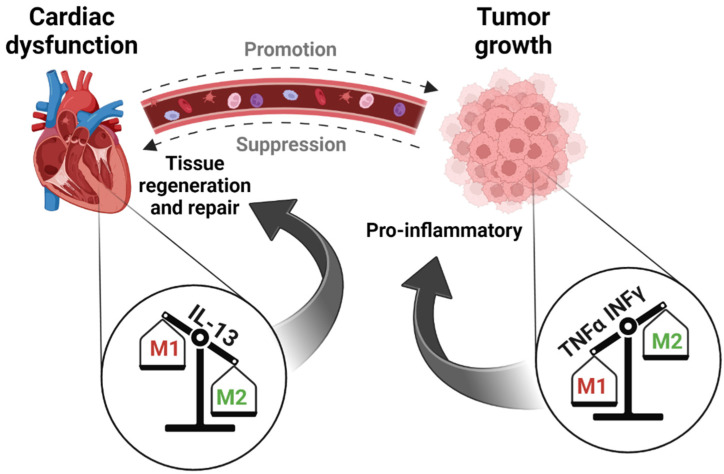
Graphical abstract describing the manuscript’s main findings. Cancer growth suppresses fibrosis, inhibits cardiac hypertrophy and ameliorates cardiac contractile function. This is achieved via an M1-to-M2 macrophage polarization in the heart, while M2-to-M1 macrophage polarization occurs in the tumor.

## Data Availability

Data and materials are available upon request from the corresponding author.

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
