# Peer review of "Tumor Growth Ameliorates Cardiac Dysfunction"

_cells, 2023, doi:10.3390/cells12141853_

Round 1
Reviewer 1 Report
The aim of this study is to highlight the cardiac remodeling following transverse aortic constriction (TAC) in the presence or absence of proliferating breast cancer cells. However, I do not recommend publication because the current proposal lacks innovation. For example, the authors mentioned that Tumor growth ameliorates cardiac dysfunction. However, several studies reported that breast cancer promotes cardiac dysfunction
Reference: Breast Cancer Promotes Cardiac Dysfunction Through Deregulation of Cardiomyocyte Ca2+-Handling Protein Expression That is Not Reversed by Exercise Training. J Am Heart Assoc. 2021 Feb;10(5):e018076. doi: 10.1161/JAHA.120.018076.
Author Response
Authors comments to reviewer #1 –
The current manuscript lacks innovation -The reviewer cited a published study describing the effect of PyMT tumors resulting deregulation of Ca+2 handling in healthy heart. While in the cited manuscript the PyMT tumor develops spontaneously, it may impose developmental changes in the heart. The cancer cells that we implanted were generated from similar transgenic mouse model, however, the cancer cells were implanted in adult mice after the heart was fully developed and therefore, display no overt developmental defects. In addition, we have performed RNA-seq analysis with RNA derived from naïve tumor bearing mice and observed no gene expression alterations what so ever. In addition, our suggested tumor beneficial effect are on a remodeled heart and no effect is observed on healthy heart.
Novelty of manuscript -The submitted manuscript, is novel, original and provocative. There is no single publication that demonstrates that tumor growth ameliorates cardiac dysfunction. We show that this cardiac function improvement is mediated by M1 to M2 macrophages switch while the adaptive immune system is not involved.
We hope that the reviewer will appreciate that this novel study will be the first publication on this subject and many other will follow from our laboratory and others and will be a novel branch in cardio-oncology discipline. From our laboratory, we have two other mouse models in which tumor growth ameliorates cardiac dysfunction. One is the Duchenne muscular dystrophy mouse model, currently in Research Square preprint DOI: 10.21203/rs.3.rs-2642165/v1. The study is cited in the current submitted manuscript. Another study that involves spontaneous cardiac remodeling using transgenic mice is currently under preparation.
Reviewer 2 Report
The authors analyzed heart remodeling after narrowing of the ascending aorta (myocyte hypertrophy, interstitial fibrosis) in mice with and without malignant tumors. They found significantly less pathological and functional changes (ECG) in the myocardium of tumor-bearing mice and proofed as a result of an indirect effect of tumor cells via IFN-gamma and M1 macrophages. Evidence for this was obtained by excluding the influence of acquired immunity by repeating the experiment on the SCID mouse and blocking mechanism of M1-M2 transition. These experimental results are exciting and encourages further research into the "beneficial" effects of tumors in the body on other organs. Contrary to this model, it is well known that poor heart function has a stimulating effect on tumor development.
Author Response
Authors comment to reviewer #2 – We thank the reviewer that finds our manuscript novel and suitable for publication in Cells.
Reviewer 3 Report
Awwad et al. investigated cardiac remodeling after TAC surgery in the presence or absence of breast or lung cancers in mice. Their main findings are that the presence of both tumors ameliorated cardiac hypertrophy and fibrosis and improved the contractile function by enhancing the M1 to M2 macrophage polarization in the heart. The authors use plenty of methods; the experiments seem to be carefully designed and performed. Their findings are interesting, however surprising and controversial to the clinical experiences.
1. It is clinically surprising that the presence of solid tumors can improve heart failure. Are there similar or contradictory literature data on your results? Can a solid tumor worsen cardiac function in healthy animals or other types of heart failure (not TAC-induced)? There is some data about colon cancer and cardiac dysfunction (https://doi.org/10.1093/cvr/cvac066.247). Please introduce the literature background better.
2. What are the similarities and differences in tumors caused by PyMT and LLC compared to human breast and lung cancers? Please explain their human clinical relevance better in the Introduction.
3. a) How many mice were used in total and per group? b) What was their body initial body weight? c) How many animals died during the study due to TAC surgery or bearing tumors? d) What was the ratio between male and female animals? e) How long were the mice followed up after TAC surgery? Please give these details in the Methods section.
4. a) Please give the ejection fraction as the gold standard of general systolic function assessment. b) You should give the wall thicknesses to show the echocardiographic signs of LVH. c) Did you measure the cardiomyocyte cross-sectional areas? d) Did you measure the diastolic parameters as well? If yes, please give them at least in Tables.
5. Breast cancer in males is a rare condition. a) What was the rationale for using both sexes in breast cancer-bearing mice? b) Were there any differences in the severity of hypertrophy, fibrosis, and heart failure between the sexes?
6. Why you did not use a sham-operated group also? What are the effects of the used tumors in healthy or sham-operated animals? Please discuss these issues.
7. a) In the qPCR methods, Hsp90 is named the housekeeping gene. In contrast, in the Figure legends, Gapdh is the housekeeping gene. b) Please expand the description of qPCR (number and temperature of cycles).
8.) Untreated breast and/or lung cancers might compensate or improve the heart failure for a short term, or the patient dies of the cancerous disease earlier as the "beneficial effects" of tumor on fibrosis and LVH are present. What is the clinically relevant take-home message of your study? How can we induce the cardiac M1 to M2 transition in heart failure without tumors? Please discuss it.
9.) Please check and renumber your reference list and the citations in the text: there are more references under No. 3-4 in the reference list.
Author Response
Authors comment to reviewer #3 – we thank the reviewer that finds our manuscript interesting, surprising and provocative.
Point-point author comments to reviewer -
- It is clinically surprising that the presence of solid tumors can improve heart failure. Are there similar or contradictory literature data on your results? Can a solid tumor worsen cardiac function in healthy animals or other types of heart failure (not TAC-induced)? There is some data about colon cancer and cardiac dysfunction (https://doi.org/10.1093/cvr/cvac066.247). Please introduce the literature background better.
Author response: This is the first study showing a beneficial effect of tumor on the diseased heart. We used breast and lung cancers. Besides, we have other mouse models in which tumor growth ameliorates cardiac dysfunction. One is the Duchenne muscular dystrophy mouse model, currently in Research Square DOI: 10.21203/rs.3.rs-2642165/v1. The study is cited in the current submitted manuscript. Another study that involves spontaneous cardiac remodeling using transgenic mice is currently under preparation.
- What are the similarities and differences in tumors caused by PyMT and LLC compared to human breast and lung cancers? Please explain their human clinical relevance better in the Introduction.
Author response: Both PyMT and LLC are considered clinically relevant models for human breast and lung cancer respectively.
- a) How many mice were used in total and per group? b) What was their body initial body weight? c) How many animals died during the study due to TAC surgery or bearing tumors? d) What was the ratio between male and female animals? e) How long were the mice followed up after TAC surgery? Please give these details in the Methods section.
Author response:
- The number of animals is indicated in the legend of each figure and in many figures each mouse is represented by a dot.
- The mouse body weight was between 20-25 gr.
- Following TAC surgery we typically have less than 10% mice death. No death was associated with tumor growth.
- Each experiment gender was unique. All PyMT experiments used females mice only. All LLC experiments male mice were used. This point was specified in the methods section.
- Mice were followed until end-point typically 3-4 weeks following TAC surgery depending on tumor size according to humane endpoint.
- a) Please give the ejection fraction as the gold standard of general systolic function assessment. b) You should give the wall thicknesses to show the echocardiographic signs of LVH. c) Did you measure the cardiomyocyte cross-sectional areas? d) Did you measure the diastolic parameters as well? If yes, please give them at least in Tables.
Author response:
a) EF% data was added to Supplemental Figure #1C
b) Wall thickness was added to supplemental Figure #1A and 1B.
c) Cross sectional area was added to supplemental Figure #1 D and 1F.
d) The diastolic parameters for all the experiments are found in the Supplementary tables 2-6.- Breast cancer in males is a rare condition. a) What was the rationale for using both sexes in breast cancer-bearing mice? b) Were there any differences in the severity of hypertrophy, fibrosis, and heart failure between the sexes?
Author response:
- As indicated in author response to comment # 3d. Breast cancer was used solely in female mice; this was also highlighted in the methods section.
- No sex difference was observed in all cardiac remodeling parameters in the timeline of the presented experiments.
- Why you did not use a sham-operated group also? What are the effects of the used tumors in healthy or sham-operated animals? Please discuss these issues.
Author response:
In our previous manuscript, we used sham control and observed no effect of tumor implantation on healthy or sham-operated animals (Avraham et al. 2020). Moreover, previous cardiac RNA sequencing analysis from our laboratory showed no effect on gene expression of cancer presence in healthy mice (Data not shown).
- a) In the qPCR methods, Hsp90 is named the housekeeping gene. In contrast, in the Figure legends, Gapdh is the housekeeping gene. b) Please expand the description of qPCR (number and temperature of cycles).
Author response:
- This was corrected in the methods section. GAPDH or Hsp90 housekeeping genes were used to normalize Heart mRNA, as indicated in the legend of each figure.
- The information was added to the qPCR methods section.
8.) Untreated breast and/or lung cancers might compensate or improve the heart failure for a short term, or the patient dies of the cancerous disease earlier as the "beneficial effects" of tumor on fibrosis and LVH are present. What is the clinically relevant take-home message of your study? How can we induce the cardiac M1 to M2 transition in heart failure without tumors? Please discuss it.
Author response:
The take home message is that harnessing tumor paradigms may have beneficial cardiac dysfunction outcome. Mimicking tumor cytokine release is suggested to ameliorates cardiac dysfunction. This message appears in the discussion and conclusions sections.
More illustration about this issue was added to the discussion.
9.) Please check and renumber your reference list and the citations in the text: there are more references under No. 3-4 in the reference list.
Author response:
References list was fixed.
Round 2
Reviewer 1 Report
The purpose of this research is to emphasize the cardiac remodeling that occurs after transverse aortic constriction (TAC) in the presence or absence of proliferating breast cancer cells. However, given the current proposal lacks novelty, I do not suggest publishing it. The authors proposed state that tumor development improves cardiac dysfunction. Several studies, however, have found that breast cancer causes cardiac dysfunction in clinical settings and research experimental animals
References:
1- Breast Cancer and the Cardiovascular Disease: A Narrative Review. Cureus. 2022 Aug 12;14(8):e27917. doi: 10.7759/cureus.27917.
2- Breast Cancer Promotes Cardiac Dysfunction Through Deregulation of Cardiomyocyte Ca2+-Handling Protein Expression That is Not Reversed by Exercise Training. J Am Heart Assoc. 2021 Feb;10(5):e018076. doi: 10.1161/JAHA.120.018076.
Reviewer 3 Report
The authors answered my questions and requests. They thoroughly revised their MS and improved its quality. Therefore, I suggest accepting their revised MS in Cells.